# Noncoding RNAs and Midbrain DA Neurons: Novel Molecular Mechanisms and Therapeutic Targets in Health and Disease

**DOI:** 10.3390/biom10091269

**Published:** 2020-09-03

**Authors:** Emilia Pascale, Giuseppina Divisato, Renata Palladino, Margherita Auriemma, Edward Faustine Ngalya, Massimiliano Caiazzo

**Affiliations:** 1Department of Molecular Medicine and Medical Biotechnology, University of Naples “Federico II”, Via Pansini 5, 80131 Naples, Italy; emilia.pascale@unina.it (E.P.); giuseppina.divisato@unina.it (G.D.); re.palladino@studenti.unina.it (R.P.); marg.auriemma@studenti.unina.it (M.A.); eddo1566@gmail.com (E.F.N.); 2Department of Pharmaceutics, Utrecht Institute for Pharmaceutical Sciences (UIPS), Utrecht University, Universiteitsweg 99, 3584 CG Utrecht, The Netherlands

**Keywords:** microRNA, long noncoding RNA, dopamine neurons, Parkinson’s disease, direct cell conversion or reprogramming, RNA therapeutics

## Abstract

Midbrain dopamine neurons have crucial functions in motor and emotional control and their degeneration leads to several neurological dysfunctions such as Parkinson’s disease, addiction, depression, schizophrenia, and others. Despite advances in the understanding of specific altered proteins and coding genes, little is known about cumulative changes in the transcriptional landscape of noncoding genes in midbrain dopamine neurons. Noncoding RNAs—specifically microRNAs and long noncoding RNAs—are emerging as crucial post-transcriptional regulators of gene expression in the brain. The identification of noncoding RNA networks underlying all stages of dopamine neuron development and plasticity is an essential step to deeply understand their physiological role and also their involvement in the etiology of dopaminergic diseases. Here, we provide an update about noncoding RNAs involved in dopaminergic development and metabolism, and the related evidence of these biomolecules for applications in potential treatments for dopaminergic neurodegeneration.

## 1. Introduction

Dopamine (DA) is one of the major neurotransmitters able to mediate primary physiological functions such as motor coordination, emotions, memory, and neuroendocrine regulation [1]. These physiological functions in our body are mediated by five subtypes of DA receptors (D_1_, D_2_, D_3_, D_4_, and D_5_). All DA receptors are widely expressed in the central nervous system (CNS) and play an important role in physiological and pathological conditions. Alterations in the DA system and its receptors in the CNS are associated with Parkinson’s disease (PD), addiction, schizophrenia, attention deficit hyperactivity disorder (ADHD), and depression; and are also linked with other neurodegenerative disorders such as Huntington disease (HD) and others that result from impaired DA receptor signaling [2]. This review focuses on the CNS dopamine system from developmental, physiological, and pathological points of view. In the interest of space, the pathological aspect of this review is mainly focused on PD. PD is the second most common neurodegenerative disease and to date, neither an effective cure nor early diagnostic tools are available that could tackle the pathologies in their early phase. PD results from the loss of DA-producing neurons in an area of the midbrain called *substantia nigra* (SN), which induce motor symptoms that become apparent when 60–80% of DA neurons are lost [3]. DA neurons degenerate during the course of the disease, a process correlated with formation of intracellular alpha-synuclein (SNCA) aggregates. Despite advances in understanding of specific altered proteins, little is known about cumulative changes in the transcriptional landscape. About 98% of the total genome is represented by noncoding elements such as noncoding RNAs (ncRNAs) [4]. Although ncRNAs have long been considered transcriptional junk RNA, recent reports reassessed their key roles in almost all steps of DA signaling, as well as differentiation and viability of DA neurons. The maintenance of a healthy and functional neuron requires finely tuned transcriptional regulation, and this regulation is largely mediated by two groups of ncRNAs: microRNAs (miRNAs) and long noncoding RNAs (lncRNAs) [5,6]. MiRNAs are an abundant class of small post-transcriptional regulatory molecules that play key roles in regulating the expression of target genes. Usually, the target sequences recognized by miRNAs are present in the 3′UTR of target mRNAs [7] and these target mRNAs are most commonly repressed by Argonaute-catalyzed cleavage and/or destabilization. Further genetic studies have highlighted that miRNAs are essential for the correct function of the CNS [8,9] (Table 1). Studies conducted on *Dicer-*mutant mice showed that the production of miRNAs is essential for the development of the midbrain DA (mDA) neurons [10]. In this complex scenario, lncRNAs are emerging as important regulators in gene expression networks in brain development. Genome-wide studies have revealed that large numbers of tissue-specific lncRNAs are enriched in brain regions and some of them are involved in neurogenesis and cellular differentiation [11,12] (Table 2). LncRNAs share several features with coding mRNAs, both are transcribed by RNA polymerase II and are further capped and spliced [13] but have lower expression levels and are involved in many different regulatory circuitries, reflecting their multifunctional role in cells [14]. These versatile functions depend on their ability to act through different mechanisms of action. LncRNAs are able to act as guides, mediating epigenetic changes by recruiting chromatin-modifying enzymes to target genes [15]. They can act as scaffolds enabling the formation of ribonuclear protein complexes involved in gene regulatory events, as well as decoys for miRNA target sites able to sequester and inactivate miRNA function [15]. Here, we review recent studies indicating crucial regulatory and functional roles for miRNA and lncRNAs in the DA phenotype and signaling, discussing their importance in CNS development and their connections to dopaminergic disorders with a main focus on PD.

## 2. Noncoding RNA Regulatory Network in DA-Neurons Development

DA neurons are capable of producing and releasing DA and they play vital roles in the regulation of voluntary movement, emotion, and memory. The development of DA neurons is an area of great interest, crucial to understanding how to generate these neurons for cell transplantation in in vivo approaches to PD. During gastrulation, precise molecular mechanisms control mDA neuron development. The combined action of morphogens and transcription factors orchestrate the specification and proliferation of mDA progenitors as well as their differentiation and survival. During early neurodevelopment at the rostral end of the embryo, the diffusible molecules such as sonic hedgehog (SHH) and WNT, bone morphogenic protein (BMP), and NODAL inhibitors induce ventral and anterior signals leading to the formation of the anterior neural tube [66,67,68]. DA neurogenesis takes place in the ventricular zone (VZ) and is also controlled by two proneural genes, *Mash1* (mouse achaete-schute homolog 1) and *Ngn2*. These two proneural transcription factors together with PITX3, LMX1b [69,70,71], OTX2, and NURR1 fine-regulate specification and maintenance of mDA specification throughout the life of an organism, thus ensuring the right progressive acquisition of DA neurotransmitter phenotype [72]. When DA progenitors divide to generate postmitotic cells that express the transcription factor NURR1, these cells migrate through the intermediate zone (IZ) while they differentiate and become tyrosine hydroxylase (TH)+, thus identifying their neuronal phenotype. Importantly, expression of the DA neurotransmitter phenotype is dependent upon NURR1, which regulates proteins critical for DA synthesis, such as TH and the DA transporter (DAT).

### 2.1. miRNA Regulation of DA Neurons

MicroRNAs are considered key regulators in the development and maintenance of DA neurons. Accumulating evidence supports how precise, time-controlled modulation of miRNAs and transcription factors in neuronal cell maturation and differentiation contribute to the establishment of specific neuronal subtypes in the CNS in vivo and in vitro [73] (Figure 1). Results deriving from the phenotypic analysis of Dicer transgenic mice models generated using a conditional DAT-Cre line revealed a pivotal role for miRNAs in CNS development and in the differentiation of different cell types [74,75]. Interestingly, *Dicer* ablation caused the degeneration of DA neurons responsible for PD-like symptoms [75]. Subsequently, *miR-let-7b* was the first miRNA identified to regulate the switch of neural stem cell (NSC) proliferation and differentiation by targeting TLX and the cell cycle regulator cyclin D_1_ [16]. The fine-tune regulation of NSC proliferation and differentiation involved also *miR-184* that has been reported to modulate the transcripts of *Numbl*, known to repress NSC proliferation and enhance their differentiation [17]. Among brain-enriched miRNAs, *miR-124* and *miR-9* have been well described in promoting differentiation of embryonic and adult NSCs and NPs [18,76,77]. Notably, miR-124 was shown to regulate neuronal differentiation by suppressing *Sox9* expression in adult NSCs and promotes differentiation of NPs by modulating a network of nervous system-specific alternative splicing by suppressing *Ptbp1* expression [18,78]. Instead, *miR-9* inhibits NSC proliferation but promotes differentiation through a feedback regulation of a nuclear receptor TLX [19]. Similarly, *miR-125* promotes differentiation of neural progenitor into neurons with dopaminergic fate [20] through a feedback regulation with Lin-28—a pluripotency factor that controls miRNA processing in NSCs [21]. A recent discovery showed that *miR-34b/c* plays a crucial role during DA neuron differentiation. Modulating WNT1 signaling—a key morphogen in the embryonic midbrain [21]—*miR-34b/c* promotes cell cycle exit and induces dopaminergic differentiation [22]. In vitro data also proved that the combination of the transcription factors ASCL1 and NURR1 with *miR-34b/c* leads to high yield of transdifferentiated fibroblast-derived DA neurons [22]. Among recently characterized miRNAs, *miR-135a2* has been described to modulate the WNT signaling during midbrain development [23]. Crucial to initiate this process are the transcription factors Lmx1b/a [66,79,80,81,82]. Mechanistically, Lmx1b-*miR135a2* feedback loop appears to be key in the downregulation of WNT1 morphogen and governing the molecular establishment of mDA progenitors [23,83]. Other miRNAs, in addition of *miR-133b*, are known to direct DA neurons differentiation [23,84,85]. Although several studies have highlighted a negative feedback loop between *miR-133b* and *Pitx3* as crucial to poise the maturation and function of DA neuron development [10,86], another miRNA, *miR-132*, has been also identified as a key player in DA neuron development. The last mentioned miRNA plays a fundamental role in the differentiation of DA neurons by directly regulating the expression of *Nurr1* (also known as nuclear receptor subfamily 4 group A member 2; Nr4a2), an important transcription factor involved in determining DA neuron fate [25]. Yang and colleagues proved that *miR-132* overexpression in ES cells dramatically inhibited the appearance of TH-positive cells by suppressing *Nurr1* expression [84]. Furthermore, it has been pointed out that *miR-132* knockdown led to the increased expression of a variety of synaptic proteins, including GLUR1 and synapsin [87]; thus, showing that *miR-132* acts as a negative regulator of synapse maturation and DA cell differentiation [88]. Novel miRNAs have been identified in human NSC differentiation regulating the dopaminergic fate such as *miR-153*, *miR-324-5p/3p*, and especially *miR-181a*. The latter was reported to have a crucial role in DA neuron development, particularly in promoting neuroepithelial-like stem cell switch from self-renewal to neuronal differentiation [20]. In addition to the role of self-renewal and differentiation modulators, the brain-specific *miR-153* is preferentially expressed in neurons [89] and it was shown to downregulate SNCA protein levels [34], which play important roles in the pathogenesis of PD; whereas the *miR-324-5p/3p* inhibits proliferation while promoting neural differentiation of murine cerebellar granule cell progenitors (GCPs) into mature granule cells by antagonizing Hedgehog signaling [90]. Another brain-enriched miRNA involved in neurodevelopment process and synaptic maturation is *miR-137*, which exhibits a critical role in regulating DAT expression, a key element in the DA signaling pathway [91]. Among the multiple effects of *miR-137* extensively investigated, several studies show that *miR-137*—whose expression is coregulated by DNA methyl-CpG-binding protein (MeCP2) and transcriptional factor *Sox2*—negatively regulates neuronal maturation of adult NSC by targeting *Ezh2*, thus promoting proliferation [27]. On the contrary, *miR-137* promotes neuronal maturation by targeting the ubiquitin ligase mind bomb 1 (*Mib1*), thus affecting the structure and function of neurons [92]. 

### 2.2. Functional Roles of Long Noncoding RNAs in DA Neurons Development

The identification of functional lncRNAs involved in controlling development and the general function of neurons is expanding [12]. A subset of the genomewide approach showed that lncRNAs exhibit higher tissue specificity and highlight a potential *cis*-regulation mechanism of lncRNAs in modulating the transcription of coding genes in the developing and adult mouse brains [93]. The transition from NSCs to neural progenitors and then to fully differentiated neurons is regulated by complex interactions between lncRNAs and other factors. The lncRNA *Rmst*, rhabdomyosarcoma 2-associated transcript, has been identified as a requirement for neuronal differentiation [44]. *Rmst* was found to be highly expressed in mDA neuronal precursors and it is also coexpressed with the midbrain transcription factor LMX1A in the developing mouse brain [94]. Previous studies also revealed that *Rmst* associates with SOX2, a cardinal transcription factor controlling NSC fate. A specific neuronal lncRNA *Pnky* is expressed in the nucleus of dividing NSCs and highly conserved during the development of the mouse and human brain. *Pnky* controls neurogenesis of ventricular–subventricular zone stem cells through the direct binding to PTBP1, a splicing factor that functions as a repressor of neuronal differentiation [45]. Currently, further reports show that another lncRNA, *TUNA*, promotes the differentiation of NSCs into glial cells [46]; similarly, *Neat1* has been shown to regulate the NSCs differentiation into oligodendrocytes [47,95]. Altogether, this evidence proves that the dynamic expression of lncRNAs in the CNS is crucial for neural cell fate determination [48,93,95]. Among lncRNAs involved in DA development [96], *Gomafu*, known also as *Miat*, has been reported to be necessary for the correct DA transmission and neurobehavioral phenotypes [49]. Accordingly, *Gomafu* knockout mice display increased DA levels in the brain with an excessive hyperactivity after exposure to the psychostimulant methamphetamine [49].

## 3. Noncoding RNA Regulatory Network in DA Neuron Physiology 

DA is a catecholamine neurotransmitter produced in neurons of both the central and peripheral nervous systems. It is stored in vesicles in axon terminals by the vesicular monoamine transporter 2 (VMAT2) and released when the neuron is depolarized. DA neurons form a neuromodulation system that originates in the SN, in the ventral tegmental area (VTA) and in the hypothalamus, and acts on G protein-coupled DA receptors to regulate all of the physiological functions in a specific manner. Based on their biochemical and structural properties, DA receptors are divided into two main groups. The D_1_-like group includes the D_1_ and D_5_ receptors; whereas the D_2_-like group consists of the D_2_, D_3_, and D_4_ receptors. The D_1_ receptor is the most abundant among the five in the CNS and regulates the development of neurons when bound by DA. D_1_- and D_2_-like receptors have high density in the striatum and SN. These receptors are essential in regulating locomotor activity, memory, and learning and are also involved in signal transduction pathways that are linked to various neuropsychiatric disorders. D_1_ and D_5_ receptors stimulate adenylyl cyclase (AC) activity and activate phospholipase C inducing intracellular calcium release. The D_2,_ D_3_ and D_4_ receptors are expressed in the brain—mainly in striatum, cerebral cortex, hippocampus, and pituitary—and play an important role in postsynaptic receptors by decreasing neuronal excitability and inhibiting DA release. The importance of the DA system in the brain is particularly relevant for PD, which is the result of degeneration of the DA neurons of *substantia nigra pars compacta* (SNc). Different neurodegenerative diseases also show increased or decreased DA release. For instance, in PD, these patients classically display upregulated expression of D_1_R and D_2_R with hypersensitive response to DA. 

### 3.1. miRNA Regulation of DA Signaling

MiRNAs have been shown to play a critical role in synaptogenesis and in the whole process associated with neuronal maturation essential for the CNS to control physiological functionality [97]. To date, growing evidence supports a key role of miRNA in regulating the DA neurotransmission and more in the synaptogenesis in general (Figure 2). The *mir-132/mir-212* cluster is also reported to mediate dendritic growth and spine formation [29]. Another brain relevant miRNA, *miR-134*, is localized to the synapto-dendritic compartment of rat hippocampal neurons. *miR-134* is reported to negatively regulate the size of dendritic spines, which are the postsynaptic sites of excitatory synaptic transmission, by targeting and repressing *Limk1* [30]. In this scenario, DA receptors display upregulated expression causing hypersensitive response to DA [98,99,100]. Importantly, DA D_1_ (*Drd1*) and D_2_ (*Drd2*) receptors are the most abundant in the striatum and their regulation is crucial for the control of many different physiological and behavioral functions in mammals [101]. Alteration of D_1_ and D_2_ receptors are some of the pathological hallmarks in PD and schizophrenia patients. In vitro data highlights regions of putative miRNA binding sites that finely control the DA receptors post-transcriptionally. One of the miRNA regulations well characterized is *miR-142-3p*. The latter one has been described to be highly expressed in the basal ganglia and prefrontal cortical region in the brain [102], and it has been demonstrated to directly interact with only a single consensus binding sites within the 3′UTR of D_1_ receptor [31], thus modulating specifically the D_1_ signaling. The inhibitor regulation of *miR-142-3p* results in an increase of cAMP production and phospho-DARPP-32 levels in the mouse catecholaminergic cell line [31]. Further scientific information showed that the *miR-15a*, *miR-15b*, and *miR-16* also inhibit *Drd1* gene expression in different human cell lines. Interestingly, all of these miRNAs specifically bind the same DNA sequence in *Drd1* 3′UTR essential for the post-transcriptional regulation [32]. Similar studies conducted in a *Drosophila* model, which recapitulates PD genetic alteration of SNCA, displayed altered *miR-137* control of D_2_ receptor [2,103]. Genomewide association study (GWAS) remarkably showed that *miR-137* dysregulation is strongly associated with the etiology of schizophrenia pathology [101]. In fact, *miR-137* enhances D_2_ receptor expression through *miR-137*-TLX-*miR-9*-D2R intracellular cascade resulting in hyperdopaminergic response typical of schizophrenia [33]. Additional scientific data show the central role of two brain-expressed miRNAs—*miR-326* and *miR-9*—regulating dysregulation of D_2_ receptors. During DA neuron differentiation, *miR-326* and *miR-9* show an opposite expression profile and such inverse correlation is indicative of a post-transcriptional regulation of *Drd2* by both miRNAs [33]. Interestingly, *miR-326* and *miR-9* also represent promising biomarkers and drug targets for the treatment of schizophrenia, involving an abnormal DA receptor function [104]. 

### 3.2. LncRNA Regulation of DA signaling 

The identification of functional lncRNAs involved in mediating the general functions of neurons and controlling the synaptic signaling is increasingly expanding. The first annotated lncRNA in the CNS was brain cytoplasmic 1 (*BC1*). *BC1* is reported as a cytoplasmic lncRNA, localizes to dendrites, and is involved in regulating the postsynaptic signaling by repressing glutamate receptor signaling (mGluR) and promoting neuronal plasticity. In vivo, loss of function of *BC1* induces upregulation of D_2_ receptor with excitability of neurons and abnormal behavior [50]. Indeed, this mouse model has been used for epilepsy modeling [105]. During the last years, research on the mechanism controlling the onset of neurodevelopmental diseases shed light on the role of specific lncRNAs—*NONHSAT089447*, *NONHSAT021545*, and *NONHSAT041499*—that play a regulatory role on the DA receptor signaling pathway [51]. Here, the authors screened these lncRNAs and found them significantly upregulated in schizophrenia patients compared to healthy controls [106], and subsequently demonstrated the tight connection between lncRNA expression and the DRD signaling pathway. Specifically, in vitro data reveals that the DA receptors DRD3 and DRD5 are reported to regulate the composition and release of DA, and their downstream signals are activated by *NONHSAT089447* expression. In fact, DA signaling was suppressed when *NONHSAT089447* expression was repressed by siRNA, while the overexpression of *NONHSAT089447* activated DRD3 and DRD5 DA receptors, respectively [106]. The latter mentioned that receptors represent possible targets for study and application in the diagnosis and treatment of schizophrenia. DAT gene (*Slc6a3*) has also recently been found to be regulated by a lncRNA transcribed in the 3′UTR of *AZI2* [52]. 

## 4. Noncoding RNAs Regulatory Network in Neurological Diseases 

In the last years, it has been becoming clear that ncRNA dysregulation plays a critical role in the etiology of human neurological disorders. Actually, ncRNAs have emerged as potentially important players in this field; even more recent studies underline their pivotal roles in several neurological disorders, among which are schizophrenia [107], addiction [108,109], and depression [110], as already summarized in these previous reviews. Here, we will specifically focus on the current advances in ncRNA research involvements in PD. Neurodegenerative diseases such as PD are the result of progressive degeneration of neurons; in the long run, leading to cognitive and functional disabilities [111]. PD is the second most common neurodegenerative disease and is associated to the degeneration of DA neurons that often correlates with an excessive deposition of the SNCA in the SN [112]. Despite intensive research, the molecular mechanisms initiating and promoting PD are still unknown. So far, ncRNAs—in particular miRNAs, lncRNAs, and more recently a newly recognized subclass of lncRNAs known as circular RNAs (circRNAs) [113], represent interesting candidates to understand the etiology of PD and its consequent progression. In fact, a recent research highlighted that ncRNAs can affect proper CNS development and also result in neurological diseases [114]. Consistently, several hundreds of ncRNAs have been underlined in brain development and in every aspect of brain function, including neurogenesis, neural differentiation and maintenance, and synaptic plasticity [12,115]. Previous work showed several links between ncRNAs and PD-related genes [42,54], and many miRNAs have been identified as potentially involved in the determination of DA neuron phenotype (i.e., *miR-133b*, *miR-218*, *miR-34b/c*) [10,22,85]. Intriguingly, it has been shown that miRNAs [116,117] and lncRNAs [118] are altered in PD patient samples. Although, an analysis on expression levels and PD stages identified dysregulated ncRNAs already in the early disease stage and during the course of PD [53], little is known about their direct or indirect impact in this context. Therefore, lncRNAs and miRNAs represent promising biomarkers targets for prognostic, diagnostic, and therapeutic applications for PD.

### 4.1. miRNA Regulation in Parkinson’s Disease

PD results from a loss of DA neurons in the SN, and SNCA accumulation is principally related to the pathogenesis of PD. Growing evidence supports specifically the involvement of miRNAs in the regulation of SNCA accumulation, responsible for the loss of DA neurons [119]. Specific miRNAs control SNCA expression. Among them, *miR-7* and *miR-153* have been mostly described in detail. They post-transcriptionally regulate *SNCA* by binding its 3′UTR, therefore suppressing its expression. In addition, the consequent downregulation of SNCA due to *miR-7* and *miR-153* protects cells from oxidative stress [34,35]. A further GWAS also revealed that SNCA expression is affected by a set of miRNAs differentially expressed in PD, such as *miR-34b/c* and *miR-214*, which directly bind its 3′ UTR [36]. As abovementioned, miRNAs represent promising biomarkers for early recognition of the onset of disease and possible therapeutic targets. Interestingly, PD patients showed reduced levels of *miR-7* in the SN of the brain, and depletion of this miRNA is functionally related with SNCA accumulation and with the further neuron loss [37]. Further investigations identified other miRNAs as well as *miR-433* that bind a single nucleotide polymorphism (SNP) in the promoter region of *Fgf20* gene causing the overexpression of SNCA [38]. Other studies on miRNA profiles in plasma samples of PD patients detected significant upregulation of *miR-331-5p* [120], as well as of *miR-20a*, *miR-16*, and *miR-320* [40]. Moreover, a common feature of many neurological diagnoses is represented by impaired synaptic transmission. One of the dysregulated miRNAs specifically related to mDA neurons in PD patients is *miR-133b*, which was reported to generate a feedback loop with *Pitx3* in controlling the proper mDA neuron differentiation [10]. Further, an in vitro study revealed that *miR-124* suppression in DA neurons increased neuronal autophagy and apoptosis by regulating the AMPK/mTOR signaling pathway in PD [41]. Research on links between ncRNAs and PD-related genes highlighted the relationships with specific miRNAs. In one of these studies, it was shown that LRRK2—one of the few genes causing familial PD [42]—negatively regulates *miR-let-7* and *miR-184* in DA neurons, leading to the defects in cell division and cell death [42]. Moreover, *miR-205* also regulates LRRK2 and has significantly lower levels in the frontal cortex and striatum of PD patients. In fact, evidence from mouse models study revealed that the downregulation of *miR-205* induces upregulation of LRRK2 protein expression [43]. To date, the biggest open challenge is to identify which miRNAs are directly associated with the causes and progression of DA neurodegeneration.

### 4.2. Long Noncoding RNA Regulation in Parkinson’s Disease

Several studies indicate that abnormal expression of lncRNAs is linked to different human neurological diseases, including PD, AD, and HD [118]. It is widely accepted that lncRNAs are highly expressed in the CNS [95] and constitute key regulators of neural development by interacting with histone modifiers, transcription factors, mRNA decay, and alternative splicing, thus modulating behavior and cognition functions [13]. Interestingly, literature reports that the differential expression profile of lncRNAs such as *H19*, *MALAT1*, *SNHG1*, and *TncRNA* occurs in the early stage of the pathological process of PD, resulting in upregulation in PD patients [53]. Aggregation of SNCA protein [121] is among the key causes of degenerations of DA neurons [122], and current studies highlighted multiple lncRNAs involved in this process. Commonly, the balance of SNCA is maintained by the combined actions of the ubiquitin-proteasome system and the lysosomal autophagy system in synergism with various lncRNAs, which represent a new regulatory layer of this process. Research findings reveal that specific lncRNAs play a protective role in PD, like *AS Uchl1*. Advances in studying the potential role of the lncRNAs in DA signaling revealed a key function for an antisense lncRNA to the *Uchl1* gene (AS-Uchl1). Ubiquitin carboxy-terminal hydrolase L1 (*Uchl1*) is a gene involved in the ubiquitin-proteasome system (UPS) of PD, responsible for removing DNA damage and preventing cell apoptosis [54]. *Uchl1* is significantly repressed in DA neurons of PD models and is regulated by NURR1 [55]. Specifically, *AS-Uchl1* is representative of a natural antisense lncRNAs family known SINEUPs (SINEB2 sequence to UP-regulate translation) that activate translation of their sense genes [123]. *AS-Uchl1* induces *Uchl1* expression by increasing its translation. Functional in vivo studies in mice and the even more recent Drosophila model showed that lack of UCHL1 resulted in PD phenotype such as motor dysfunction, instability of ubiquitin level, and exhibit DA neuron degeneration in MPTP-treated conditions [55,124]. For this reason, manipulation of *Uchl1* expression has been proposed as a tool for therapeutic intervention [55]. Similarly, it has been shown that the lncRNA *NEAT1*, which is overexpressed in the SN of PD, plays a neuroprotective role against drug-induced oxidative stress. Moreover, in a rat model of PD, the downregulation of the lncRNA *UCA1* inhibits the PI3K/Akt signaling pathway, with a resultant reduction in the damage of the DA neurons, as well as the oxidative stress and inflammatory response associated with PD [57]. Based on these studies, it is arguable that *UCA1* might be considered a novel target for therapeutic intervention of PD. Other lncRNAs have been demonstrated to play important roles in the apoptosis of DA neurons, processes closely related to mitochondrial dysfunction and oxidative stress correlated to PD disorder. In particular, the lncRNA *HOTAIR* has been shown to affect the progression of PD [125]. Indeed, the in vivo knockdown of *HOTAIR* reduces the number of SNCA-positive cells and reduces apoptosis of DA neurons [58]. Further data showed that *HOTAIR* was upregulated in PD mouse model and improved the stability of LRRK2 mRNA by enhancing its expression and thus promoting the apoptosis of dopaminergic neurons [126]. Recently, it was also reported that the inhibition of lncRNA *MALAT1* in PD mice induced the apoptosis of DA neurons by upregulating *miR-124* [59,60]. Several studies have explored the potential of lncRNAs as attractive diagnostic and prognostic factors in neurodegenerative disease [127]. For example, increasing data indicated that an increase in *Uchl1* expression could be advantageous in neurodegenerative diseases, the application of *AS-Uchl1* as an RNA-target drug may be considered a novel therapeutic tool. More in general, the therapy based on lncRNAs as a biomarker and possible therapeutic target for PD—although it is in its infancy—holds great promises for the treatment of neurodegenerative diseases. 

### 4.3. Circular RNAs as Parkinson’s Disease Biomarkers

Within the family of lncRNAs, circular RNAs (circRNAs) [128,129] have been recognized for playing an important role in pathological mammalian brain functions. CircRNAs share a circular RNA structure and are highly abundant in the mammalian brain, therefore emerging as new molecular players in disorders of the CNS including PD [61]. Several studies report that circRNAs regulate gene and protein expressions by acting as miRNA sponge. The circRNA *ciRS-7*, also known as *CDR1as*, is specifically expressed in the mammalian brain [62] and contains multiple conserved *miR-7* target sites suggesting that *ciRS-7* act as a sponge for *miR-7* and can therefore regulate [63,130] the stability of several mRNA targets in the brain through the binding to *miR-7* [62,64]. Scientific reports show that ciRS-7 negatively regulates *miR-7*, whereby *miR-7* is a direct inhibitor of SNCA protein with a crucial role in PD [131,132]. Furthermore, *circSNCA*, another circRNA, can sponge *miR-7*, thereby regulating expression of SNCA and thus resulting in decreased autophagy and increased apoptosis cells [65]. Due to its conservation, abundance, tissue-specific expression, and roles in disease progression [133], *ciRS-7* is a promising guide for the development of new diagnostic and therapeutic strategies for the prevention of neurodegenerative disorders including PD in a near future [134] even though the reliability and security of using circRNAs as a therapeutic tool need to be further investigated due to their complex roles.

## 5. Conclusions and Future Perspectives

To date, several investigations have demonstrated that ncRNAs play critical roles in the development of DA neurons and in the pathogenesis of neurodegenerative disorders. The relationship between miRNAs and human neurological diseases still necessitates full assessment. As a result of the fast development of the miRNA synthesis and release techniques, the hope of using these biomolecules as diagnostic biomarkers and as a new therapeutic strategy for neurological disorders is becoming a realistic chance (Figure 3). At the same time, the functional analysis of lncRNAs in neural development and in disease conditions remains an exciting current research topic. 

On the other hand, ncRNAs could also have an impact on other PD therapeutic approaches such as cell replacement therapy. Indeed, it has been recently shown that both mouse and human fibroblasts can be reprogrammed into functional DA neurons [135,136] and they can improve the motor symptoms of a PD model after brain transplantation [137,138,139]. Moreover, an alternative therapeutic strategy for PD can be represented by direct conversion of astrocytes into DA neurons. Using the combination of specific transcription factors and miRNA, such as NEUROD1, ASCL1 and LMX1A, and *miR-218*, it was possible to reprogram human astrocytes (in vitro) and mouse astrocytes (in vivo) into induced DA neurons [140]. Given the plasticity of astrocytes cells, trans-differentiation strategy of these cells in functional neurons hold great promise to replace lost DA neurons in PD [141]. A recent remarkable discovery by Qian and colleagues demonstrated an efficient one-step conversion of mouse and human astrocytes to DA neurons by silencing the RNA-binding protein PTBP1. In more detail, the modulation of PTB protein and its neuronal analogue nPTB activity through sequential downregulation of these factors induces the astrocyte conversion into functional neurons from human fibroblasts [142]. More interestingly, they also demonstrated that PTB1 silencing strategy is able to regenerate part of the striatal DA neurons, thus recovering motor deficits. Notably, Qian and colleagues also evidenced the efficiency of using antisense oligonucleotides against PTB (PTB-ASO) to convert in vivo neurons with functional neurophysiological properties in PD model [143]. Similar advances in the new frontiers of cell replacement therapy highlighted the possibility to efficiently convert glial cells into retinal ganglion cells (RGCs) by downregulation of PTBP1, through in vivo delivery of viral RNA-targeting CRISPR-CasRx [144]. More interestingly, this approach is able to induce neurons with dopaminergic features in the SN and relieve motor dysfunction associated with a PD mouse model [145]. Thus, glia-to-neuron conversion based on RNA editing of *Ptbp1* represents a promising in vivo approach to treat genetic diseases accompanied by neurodegeneration. Taken together, these data look very promising and suggest that the downregulation of PTB1, and use of the relative ASO, can be a potential therapy for PD [146]. This approach appears to be very promising; indeed, ASO-based therapies are becoming a realistic therapeutic strategy for the treatment of CNS [147] and have already been successfully tested in clinical trials for neuromuscular diseases, including spinal muscular atrophy (Spinraza) and Duchenne muscular dystrophy (Exondys). A similar approach is under development for HD as shown by recent promising results of a clinical trial with ASO to target mutant huntingtin transcripts [ [148],[149]] and ongoing are the trials for LRRK2-targeted ASO for PD (NCT03976349).

Overall, these latest developments in basic, translational, and clinical research show that RNA therapeutics is a valuable tool for the treatment of neurodegenerative diseases. On the other side, we here recapitulated the multiple evidences that link DA neurons to ncRNAs, therefore paving the way to combine RNA therapeutics together with gene editing and cell reprogramming [150] in the fight against PD. 

## Figures and Tables

**Figure 1 biomolecules-10-01269-f001:**
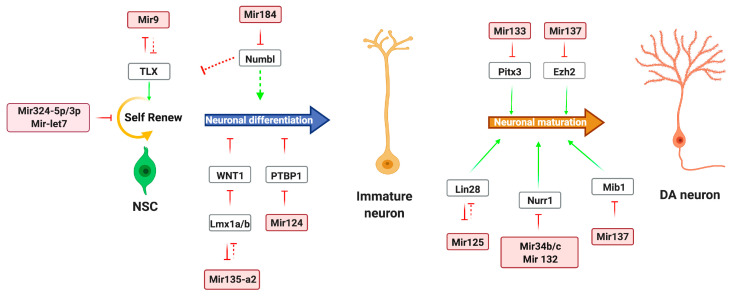
miRNA network involved in DA neuron development. Representative miRNAs involved in the control of self-renewal and proliferation from the neural stem cell (NSC) stage towards dopamine (DA) neuron differentiation. Red arrow-heads or flat-heads indicate positive or negative direct modulation of the indicated genes, respectively. Green arrows indicate overall modulation of a biological process. Dashed lines indicate hypothetical direct or indirect modulation. miRNAs are highlighted by colored boxes.

**Figure 2 biomolecules-10-01269-f002:**
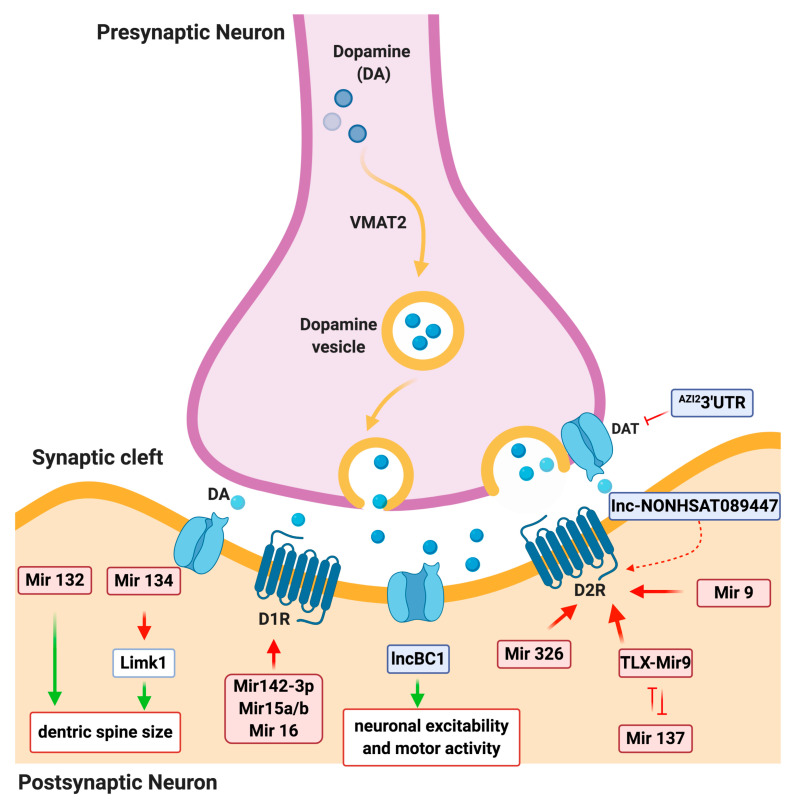
ncRNA network involved in DA neuron signaling. The picture represents the main microRNA and long noncoding RNAs involved in the regulation of dopamine (DA) neuron signaling, VMAT2 (vesicular monoamine transporter 2), and DAT (dopamine transporter). Red arrow-heads or flat-heads indicate positive or negative direct modulation of the indicated genes, respectively. Green arrows indicate overall modulation of a biological process. Dashed lines indicate hypothetical direct or indirect modulation. ncRNAs are highlighted by colored boxes.

**Figure 3 biomolecules-10-01269-f003:**
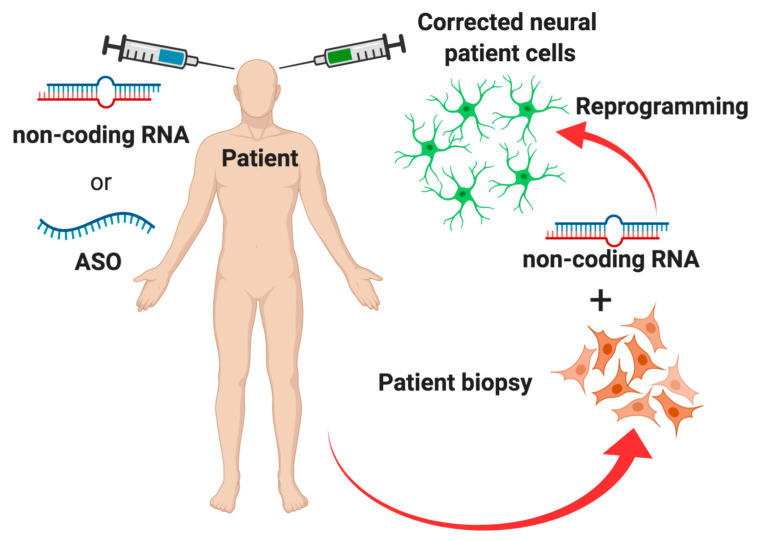
RNA therapeutics approach for neurodegenerative disease. The picture shows an ideal representation of potential RNA therapeutics approaches in which noncoding RNAs (ncRNAs) or antisense oligonucleotides (ASOs) are used to modulate genes involved in neurodegeneration. On the right side of the picture is a potential use of ncRNAs to reprogram and correct patient somatic cells. Red cells indicate diseased cells derived from a neurodegenerative disease patient, whereas green cells represent cells corrected with ncRNAs and reprogrammed towards neural phenotype. On the left side of the picture is a potential direct in vivo reprogramming/correction approach of resident neural cells.

**Table 1 biomolecules-10-01269-t001:** Summary of microRNAs (miRNAs) involved in dopamine (DA) signaling pathway.

miRNA Subgroups	Name	Function	References
miRNAs in development	miR-let-7b	Regulates neural stem cell (NSC) proliferation and differentiation	[16]
	miR-184	Binds Numbl transcript	[17]
	miR-124	Suppresses Sox9 expression, promotes differentiation of NPs	[18]
	miR-9	Inhibits NSC proliferation, promotes differentiation	[19]
	miR-125	Differentiation of neural progenitors	[20,21]
	miR-34b/c	Modulates Wnt1 signaling, promotes cell cycle exit, and induces dopaminergic differentiation	[22]
	miR135a2	Modulates Wnt1/Wnt morphogen signaling	[23]
	miR-133b	Maturation and function of DA neuron development	[10,24]
	miR-132	Differentiation of DA neurons.	[25,26]
	miR-181a	Promotes neuroepithelial-like stem cell switch from self-renewal to neuronal differentiation	[20]
	miR-137	Negatively regulates neuronal maturation of adult NSC proliferation and cell fate determination	[27,28]
miRNAs in physiology (DA signaling network)	miR-132/ miR-212 cluster	Mediates dendritic growth and spine formation	[29]
	miR-134	Negatively regulates the size of dendritic spines	[30]
	miR-142-3p	Modulates the D_1_ signaling	[31]
	miRNA-15a, miRNA-15b, and miRNA 16	Inhibit the DRD1 gene expression	[32]
	miR-137	Enhances D_2_ receptor expression	[33]
	miR-326 and miR-9	Post-transcriptional regulation of DRD2 by both microRNAs	[33]
miRNAs in neurological diseases	miR-7 and miR-153	Regulate post-transcriptionally α-synuclein	[34,35]
	miR34b/c, and miR-214	Bind directly the 3′ UTR of alpha-synuclein.	[36]
	miR-7	Its depletion is related with alpha-synuclein accumulation and with neuron loss	[37]
	miR-433	Causes the overexpression of alpha-synuclein (SNCA)	[38]
	miR-331-5p	Upregulated miRNA in Parkinson’s disease (PD) patients	[39]
	miR-20a, miR-16, and miR-320	Specifically altered in PD patients	[40]
	miR- 133b	Controlling midbrain DA (mDA) neuron differentiation	[10]
	miR-124	Increases neuronal autophagy and apoptosis	[41]
	let-7 and mir-184	Linked to defects in cell division and cell death.	[42]
	miR-205	Upregulation of LRRK2 protein expression	[43]

**Table 2 biomolecules-10-01269-t002:** Summary of long noncoding RNAs (lncRNAs) involved in DA signaling pathway in health and disease.

Long Noncoding RNA Subgroups	Name	Function	References
Long noncoding RNAs in development	RMST	Neuronal differentiation	[44]
	Pnky	Controls neurogenesis of ventricular–subventricular zone stem cells	[45]
	TUNA	Promotes the differentiation of NSCs into glial cells	[46]
	NEAT1	Regulates the NSCs differentiation into oligodendrocytes	[47,48]
	Gomafu (known also as MIAT)	Modulates dopaminergic transmission and neurobehavioral phenotypes	[49]
Long noncoding RNAs in physiology (DA signaling network)	BC1	Regulates the postsynaptic signaling	[50]
	NONHSAT089447, NONHSAT021545, and NONHSAT041499	Regulatory role on the DA receptors signaling pathway, upregulated in schizophrenic patients	[51]
	^AZI2^3′UTR	Transcriptional regulation of human SLC6A3 (DAT) and a crucial risk factor for substance abuse disorders	[52]
Long noncoding RNAs in neurological diseases	H19, MALAT1, SNHG1, and TncRNA	Are upregulated in PD patients	[53]
	Uchl1	Is responsible to remove DNA damage and prevents cell apoptosis	[54]
	UCHL1-AS (Antisense transcript of UCHL1)	Promotes translation and expression of UCHL1 which is strongly down regulated in neurochemical models of PD in vitro and in vivo	[55,56]
	NEAT1	Overexpressed in the substantia nigra of PD. Neuroprotective role against drug-induced oxidative stress.	[57]
	UCA1	Inhibits the PI3K/Akt signaling pathway	[57]
	HOTAIR	Affects the progression of PD	[58]
	MALAT	In PD mice induces apoptosis of DA neurons.	[59,60]
	ciRS-7 (CDR1as)	Negatively regulates miR-7 activities.	[61,62,63,64]
	circSNCA	Act as a sponge for miR-7 regulating alpha-synuclein expression.	[65]

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
