# Peer review of "Noncoding RNAs and Midbrain DA Neurons: Novel Molecular Mechanisms and Therapeutic Targets in Health and Disease"

_biomolecules, 2020, doi:10.3390/biom10091269_

Round 1

Reviewer 1 Report

This review is overall well written. The authors could improve this manuscript to make contents in each figures easier to understand by circumstantially explaining in figure legends. This reviewer feels that contents in legend are insufficiency. The authors should improve legends for readers to be easy to follow and catch up reviewed content.

Author Response

We are grateful to Reviewer1# for the positive comments.

As requested by Rewiever#1 we explained more in details the figures by expanding the relative figure legends.

Reviewer 2 Report

Non-coding RNA regulation is an emerging field and interesting to the biomedical research. I have few main concerns.

1) how this review is different from others such as PMID: 32721662 (there are at least six PD-related reviews)? This should be clarified in the beginning.

2) Missing refs is noticed, such as PMID: 28983843 for DAT gene, which should be included in the Fig 2.

3) Addiction is a main disease related to dopamine and there is new literature, which should be included in any new review (e.g., three GEO databases GSE112370. GSE29261, GSE21901 and many published papers; by entering "addiction" and "miRNA", I got >200 items in pubmed).

4) Circular RNA is also in the literature and database, and should be included in new review.

I think that including these areas can enhance the value of this review. for Biomolecules.

Author Response

We are grateful to Reviewer2# for the positive comments.

His points were addressed as follows:

1) As the Reviewer points out there are several reviews that describe the links between Parkinson’s disease and non-coding RNAs. On the other side our work is focused in a broader way on the link between dopamine neurons and non-coding RNAs, including a broad overview of development and signaling processes.

2) As requested we added a DAT specific reference in figure 2 legend.

3) We agree with the Reviewer that addiction is a main disease related with dopamine neurons, but in the interest of space limits we cannot cover in detail all the dopamine neuron diseases here, so we added new references of reviews that already covered the links between dopamine neurons and addiction, schizophrenia and depression.

4) As requested by the reviewer we now added a new paragraph (4.2.1) covering the link between circular RNAs and dopamine neurons.

Round 2

Reviewer 2 Report

I think the authors have addressed my comments partl, see my further questions for each point

1) As the Reviewer points out there are several reviews that describe the links between Parkinson’s disease and non-coding RNAs. On the other side our work is focused in a broader way on the link between dopamine neurons and non-coding RNAs, including a broad overview of development and signaling processes.

Where is the clarification in the revision? I don't see it in introduction and discussion

2) As requested we added a DAT specific reference in figure 2 legend.

This addressing is incomplete too: AZI2 3'UTR should be added to figure 2 diagram; the reference 55 is not correctly cited (wrong information)

3) We agree with the Reviewer that addiction is a main disease related with dopamine neurons, but in the interest of space limits we cannot cover in detail all the dopamine neuron diseases here, so we added new references of reviews that already covered the links between dopamine neurons and addiction, schizophrenia and depression.

This information should be included and explained in the text too (at least introduction and/or discussion (I see addiction is mentioned in abstract but not in the rest of the text).

4) As requested by the reviewer we now added a new paragraph (4.2.1) covering the link between circular RNAs and dopamine neurons.

Thank you. Grammar needs clean up (e.g., are -> is)

Author Response

We addressed Reviewer#2 comments as follows:

1) Where is the clarification in the revision? I don't see it in introduction and discussion.

We now added a justification for the main focus on Parkinson's disease in the review

2) This addressing is incomplete too: AZI2 3'UTR should be added to figure 2 diagram; the reference 55 is not correctly cited (wrong information)

We now added the reference in the text and adjusted the relative figure.

3)This information should be included and explained in the text too (at least introduction and/or discussion (I see addiction is mentioned in abstract but not in the rest of the text).

We now added the information in the introduction.

4) Thank you. Grammar needs clean up (e.g., are -> is)

We corrected the grammar mistakes in the text.